# *Osmunda japonica* Extract Suppresses Pro-Inflammatory Cytokines by Downregulating NF-κB Activation in Periodontal Ligament Fibroblasts Infected with Oral Pathogenic Bacteria

**DOI:** 10.3390/ijms21072453

**Published:** 2020-04-01

**Authors:** Jihyoun Seong, Jinkyung Lee, Yun Kyong Lim, Weon-Jong Yoon, Seunggon Jung, Joong-Ki Kook, Tae-Hoon Lee

**Affiliations:** 1Department of Oral Biochemistry, Dental Science Research Institute, School of Dentistry, Chonnam National University, Gwangju 61186, Korea; jhseong@jnu.ac.kr (J.S.); wlsrud1945@naver.com (J.L.); 2Department of Oral Biochemistry, School of Dentistry, Chosun University, Gwangju 61452, Korea; dbsruddl77@hanmail.net (Y.K.L.); jkkook@chosun.ac.kr (J.-K.K.); 3Jeju Biodiversity Research Institute (JBRI), Jeju Technopark (JTP), Jeju 63608, Korea; yyjkl@jejutp.or.kr; 4Department of Oral & Maxillofacial Surgery, School of Dentistry, Chonnam National University, Gwangju 61186, Korea; seunggon.jung@chonnam.ac.kr; 5Department of Molecular Medicine (BK21plus), Chonnam National University Graduate School, Gwangju 61186, Korea

**Keywords:** *Fusobacterium nucleatum*, *Porphyromonas gingivalis*, periodontal ligament fibroblast, *Osmunda japonica*, pro-inflammatory cytokines

## Abstract

Periodontal diseases are caused by bacterial infection and may progress to chronic dental disease; severe inflammation may result in bone loss. Therefore, it is necessary to prevent bacterial infection or control inflammation. Periodontal ligament fibroblasts (PDLFs) are responsible for the maintenance of tissue integrity and immune and inflammatory events in periodontal diseases. The formation of bacterial complexes by *Fusobacterium nucleatum* and *Porphyromonas gingivalis* is crucial in the pathogenesis of periodontal disease. *F. nucleatum* is a facultative anaerobic species, considered to be a key mediator of dental plaque maturation and aggregation of other oral bacteria. *P. gingivalis* is an obligate anaerobic species that induces gingival inflammation by secreting virulence factors. In this study, we investigated whether *Osmunda japonica* extract exerted anti-inflammatory effects in primary PDLFs stimulated by oral pathogens. PDLFs were stimulated with *F. nucleatum* or *P. gingivalis*. We showed that pro-inflammatory cytokine (IL-6 and IL-8) expression was induced by LPS or bacterial infection but decreased by treatment with *O. japonica* extract following bacterial infection. We found that the activation of NF-κB, a transcription factor for pro-inflammatory cytokines, was modulated by *O. japonica* extract. Thus, *O. japonica* extract has immunomodulatory activity that can be harnessed to control inflammation.

## 1. Introduction

Periodontitis is a common chronic inflammatory disease that causes the destruction of the periodontium, including the periodontal ligament (PDL), and the alveolar bone. Untreated periodontitis can lead to tooth loss and impaired mastication [1]. It is estimated that two-thirds of adults have periodontal inflammation and that the incidence of severe periodontitis in adults is approximately 10%. The severe stage of the disease is associated with a risk of certain systemic conditions, such as atherosclerosis, diabetes, and cancer [2,3,4].

Periodontal ligaments are the connective tissues located between the cementum of the teeth and the alveolar bone of the mandible or maxilla. PDL fibroblasts (PDLF) are considered to play an integral role in the maintenance and regeneration of periodontal tissue through their production of osteoblast-related extracellular matrix proteins [5]. PDLFs are known to be sensitive to lipopolysaccharide (LPS) and other pathogenic factors [6]. These inflammatory signals are attributed to an imbalanced oral microbiome composed predominantly of *Porphyromonas gingivalis, Treponema denticola, Tannerella forsythia*, and *Fusobacteium nucleatum* [7,8]. The bacterial infections and their components, including LPS, mediate the excessive secretion of pro-inflammatory cytokines from PDLFs [9,10]. Interleukin-6 (IL-6) and interleukin-8 (IL-8) are the prominent pro-inflammatory cytokines that are closely related to periodontitis, which leads to connective tissue damage and alveolar bone loss [11,12].

The current therapy for periodontitis is the mechanical removal of the pathogenic biofilm; however, this is ineffective [13,14,15]. Many studies have therefore aimed to identify new therapeutics to modulate inflammation. However, there are concerns regarding the toxicity and adverse effects of synthetic medicines [16]. Accordingly, natural products have been suggested as alternatives to synthetic drugs as they are relatively safe and inexpensive. The potential adverse effects and toxicity of plant-derived natural products also need to be elucidated, although they are generally considered to be safe and effective [17,18,19].

*Osmunda japonica* Thunb., a fern belonging to the family Osmundaceae, is native to eastern Asia. It has been used as traditional medicine for reducing fever, relieving pain, and clearing infection. For these reasons, there have been many studies of the anti-infectious properties and anti-oxidative stress effect of *O. japonica* [20,21,22]. Despite the extensive literature detailing the beneficial effects and underlying mechanisms, the effect of *O. japonica* extract on periodontal disease has not been investigated. In this study, we investigated the biological activities of *O. japonica* extract (*OJE*) on periodontal disease induced by oral pathogens.

## 2. Results

### 2.1. Suppression of O. japonica Extract on LPS-Induced Pro-Inflammatory Cytokines

We aimed to discover an effective candidate for periodontitis from natural products known to have anti-inflammatory or antibacterial effects. Human primary PDLFs were treated with whole extracts of *Malus sieboldii*, *Melia azedarach*, *Lonicera japonica*, and *Osmunda japonica*, and the cells were stimulated with LPS from *Escherichia coli* (ecLPS). Previously, we reported that 1–10 µg/mL ecLPS effectively induced inflammation in PDLFs [23]. Therefore, we determined the anti-inflammatory effects of the extracts by measuring the mRNA and protein expression of IL-6 and IL-8 after stimulation with 1 µg/mL ecLPS. Of the tested natural products, *O. japonica* extract (*OJE*) exerted a suppressive effect on the LPS-induced expression of IL-6 and IL-8 mRNA, with an approximate decrease of 63.4% in IL-6 mRNA and 83% in IL-8 mRNA induced by ecLPS in PDLFs pre-treated with *OJE* (Figure 1A,B). We also examined pro-inflammatory cytokine production in the presence of various concentrations of *OJE* (10–100 µg/mL). The extract dose-dependently reduced IL-6 and IL-8 (Figure 1B–D). We found that 100 µg/mL *OJE* significantly reduced the IL-6 and IL-8 expression after ecLPS stimulation (Figure 1).

### 2.2. Inhibitory Effect of O. japonica Extract on Oral Bacteria Induced Pro-Inflammatory Cytokines

Next, we determined whether the extract suppressed the expression of pro-inflammatory cytokines induced by bacterial infection. First, we determined the response of infected PDLFs to different doses of *OJE* (Figure 2A–F). When the PDLFs were pre-treated with 10–100 µg/mL *OJE* and stimulated by bacteria for 12 h, the expression of the cytokines was gradually decreased at the messenger level. Over 90% of IL-6 mRNA induced by *F. nucleatum* or a mixture of *F. nucleatum/P. gingivalis* was decreased in the 100 µg/mL *OJE* treatment group (Figure 2A,C). In the *P. gingivalis* group, the expression of IL-6 mRNA was restored to the level of the uninfected sample, although a dramatic increase was not induced by infection (Figure 2B). The same trends were seen for IL-8 mRNA (Figure 2D–F). In addition, we observed the secreted cytokines following oral bacterial infection (Figure 3A–F). The PDLFs were pre-treated with 100 µg/mL *OJE* and then stimulated with *P. gingivalis* or *F. nucleatum*. For the bacterial infection, the host cells were stimulated with single or mixed strains and incubated for the indicated times (0.5, 1, 4, 12, and 24 h). 

*OJE* pre-treatment significantly reduced IL-6 (Figure 3A–C) and IL-8 (Figure 3D–F) secretion in PDLFs infected with both single and mixed bacteria. The *P. gingivalis* infection induced relatively lower cytokine expression than *F. nucleatum*, but clearly induced inflammation after a 12-h incubation period. Compared with *F. nucleatum* infection (at 12 h, without OJE), the expression of IL-6 and IL-8 in *P. gingivalis*-infected cells was 90% and 82% lower, respectively (Figure 3A,B,D,E). Oral pathogens form a ‘red complex’ that consists of various oral bacteria and is found in deep periodontal pockets; this is strongly related to the progression of periodontal disease [24,25,26]. In addition, we reported that the dominant growth of *F. nucleatum* allowed *P. gingivalis* to attach to periodontal tissues by inducing NOX1/2 activation [27]. Therefore, we mimicked the red complex by mixing *P. gingivalis* and *F. nucleatum*. Infection with the mixture induced similar expression of IL-6 and IL-8 mRNA (Figure 2C,F) and protein (Figure 3C,F) to infection with *F. nucleatum* alone. *OJE* also effectively suppressed the cytokine production induced by infection with the mixture.

### 2.3. Mechanism of Immunomodulatory Effects

To investigate the effect of the extract on intracellular signaling pathway, we measured the activation of NF-κB (Figure 4A,B) and MAPKs (Appendix A). The PDLFs were stimulated with purified ecLPS and treated with single strains of *F. nucleatum* and *P. gingivalis* or their mixture. When the PDLFs were pre-treated with *OJE*, the phosphorylation of NF-κB was apparently reduced compared with the untreated control, and the activation of NF-κB in LPS-stimulated and bacteria-infected PDLFs tended to be suppressed. In our infection model, *F. nucleatum* induced a high expression of cytokines and virulence factors including LPS, adhesins, and growth factors. It should be emphasized that the LPS was an enterobacterium-type and has been found to possess biological activities comparable with those of *E. coli* [28]. Collectively, we assumed that the inflammation could be induced via toll-like receptor 4 (TLR4). To focus on TLR4-mediated cytokine secretion and as all LPS chemotypes are described to activate IRF3 as well as NF-κB, we tried to measure IRF3 activation. However, IRF3 phosphorylation was not changed throughout the experiment; indeed levels of phospho-ERK and phospho-p38 MAPKs were not different in *OJE* pre-treated and untreated PDLFs (Appendix A). In contrast to NF-κB activation, IRF3 activation is promoted by internalizing the TLR4 complex. In our experiment, the bacteria could not be introduced into PDLFs when MOI was 100 (data not shown). When the MOI was increased to 200, the bacteria could internalize the cells (Figure 4C). These data also indicated that *OJE* did not affect bacterial invasive capability.

### 2.4. Anti-Inflammatory Effect of Fractions of OJE

To report the effective chemical compounds contained in the extract, butanol, hexane, water, CH_2_Cl_2_, and ethyl acetate fractions were partitioned as described in Materials and Methods section. First, each fraction was treated to PDLFs to determine the cytotoxic effects. The ethyl acetate fraction was cytotoxic at a concentrations over 100 µg/mL (Figure 5A). Therefore, PDLFs were treated with 50 µg/mL of each fraction and stimulated with ecLPS to evaluate the anti-inflammatory effect of each fraction. The CH_2_Cl_2_ and ethyl acetate fractions dramatically decreased the expression of IL-6 and IL-8 mRNA (Figure 5B,C). At the mRNA level, the ethyl acetate and CH_2_Cl_2_ fractions showed the best effect on the reduction of IL-6 and IL-8 mRNA expression (Figure 5B,C). Therefore, we chose to examine the protein expression of these two fractions (Figure 5D,E). Again, we treated PDLFs with each of the fractions and infected the cells with single strains of *F. nucleatum* and *P. gingivalis*, and their mixture. The EtOAc and CH_2_Cl_2_ fractions strongly reduced inflammatory cytokines induced by bacterial infection (Figure 5D,E). There are several reports that *O. japonica* contains flavonoids, esters, steroids, anthraquinones, ketones, phenols, and many other similar compounds. In particular, the ethyl acetate fraction of *O. japonica* rhizome was shown to contain phenolic compounds with immunomodulatory activities [20,21,22]. In the reports, aldehyde-type phenolic compounds were shown to enhance NO production, interferon-γ, tumor necrosis-α, and interleukin-1β, whereas acetone-type phenolics inhibited the production of NO, prostaglandin E_2_, and immune mediators, including TNF-α, IL-1β, and IL-6. Although we did not purify specific compounds from the extract, it is shown that *OJE* exerts anti-inflammatory effects in periodontitis.

## 3. Discussion

Growth of specific bacteria in biofilms in an interdependent manner on the tooth surface is a common factor in periodontal diseases, including periodontitis and gingivitis [29,30]. The oral bacteria on the biofilm produce several toxic substances, such as endotoxins, mucinous peptides, fatty acids, organic acids, hydrogen sulfide, and leukotoxins [31]. In response to those stimulants, pro-inflammatory cytokines are released from oral tissues, which then recruit immune cells. However, excessive inflammation results in tissue damage and bone resorption [32]. Therefore, the modulation of inflammation should be an effective therapeutic action that inhibits disease progression. 

The loss of PDL and alveolar bone of the periodontal tissue is characteristic of periodontal diseases caused by bacterial infection. Many bacterial species are known to be associated with periodontal disease. Therefore, it is important to limit the chances of bacterial invasion and infection. In this study, we first identified the greater effect of crude *OJE* on the reduction of pro-inflammatory cytokines in PDLFs after stimulation with *E. coli* LPS compared with other natural extracts. Relative to the untreated control group, IL-6 and IL-8 mRNA expression was reduced by 63% and 81%, respectively, after *OJE* treatment (Figure 1 and Figure 2A). These results indicated that *OJE* may suppress TLR2/4-mediated cytokine production in PDLFs. Thus, we determined the effect of the *OJE* on production of the cytokines in response to bacterial infection. In a previous report, our group found that *F. nucleatum* likely interacted with gingival fibroblasts in the connective tissue and that the bacteria provided a favorable environment for other anaerobic bacteria, such as *P. gingivalis* [27,33]. *F. nucleatum* is an orange complex bacterium, and its prevalence is significantly associated with increasing pocket depth [34]. The bacteria in the orange complex are less pathogenic than those in the red complex. However, *F. nucleatum* is a strong inducer of chemokines and pro-inflammatory factors [35]. The data presented in Figure 2B show that the *F. nucleatum* infection induced higher production of the cytokines than *P. gingivalis*. The strains used in this report effectively induced IL-6 and IL-8 release, and these cytokines were markedly suppressed by *OJE* treatment (Figure 2B).

Numerous studies have shown that red complex bacteria, including *P. gingivalis*, *T. denticola*, and *T. forsythia*, increase the virulence when they are present with orange complex bacteria [24]. We also observed the extent of IL-8 secretion during the co-infection with *F. nucleatum* and *P. gingivalis*. In the co-infection group, IL-8 secretion was increased approximately 3.88-fold compared with stimulation with *P. gingivalis* alone and increased approximately 1.3-fold compared with stimulation with *F. nucleatum* alone. IL-6 secretion was not affected by *F. nucleatum* infection alone (23.1 ng/mL) or co-infection (21.39 ng/mL), but by *P. gingivalis* infection alone (Figure 2B). The discrepancy of the amount of secreted IL-6 and IL-8 may be caused by the relatively high IL-6 expression that already reached the critical point following *F. nucleatum* infection alone. After treatment with *OJE*, the bacteria-induced cytokine expression was significantly reduced at both the messenger and protein level (Figure 2A,B).

Through bacterial infection or stimulation of human primary oral cells with their components, cells secrete various inflammatory mediators, such as IL-6 and IL-8. In response to host injury and bacterial infection, IL-6 is produced and has a prominent role as an immunological mediator; however, it is known that there is excess production of this cytokine in patients with periodontitis compared with healthy individuals. IL-6 is secreted after the healthy gingival fibroblasts are stimulated with oral pathogens [27,33]. IL-8 has a role in maintaining a healthy periodontium owing to its chemotactic activity, which facilitates the infiltration of monocytes into periodontal tissues [34]. Moreover, the signaling pathway stimulated by *F. nucleatum* or other oral bacteria relies on TLR4 [36,37,38], which subsequently causes downstream activation of NF-κB [39]. Thus, this leads to the transcription of pro-inflammatory cytokines, such as IL-1β, IL-6, and IL-8, ultimately causing tissue destruction. In the current study, we showed that ecLPS stimulation and bacterial infection induced the phosphorylation of NF-κB, but not MAPKs (P38, ERK) and IRF3. Again, *OJE* treatment reduced the activation of NF-κB (Figure 4). In the data, the co-infected cells showed reduced NF-κB activation at early phase, but the phosphorylation was restored 30 min after the infection. This may be a synergetic effect of the orange and red complexes that could not be overcome by 100 µg/mL *OJE*. In addition, we did not detect any elevation of IRF3 or changes in MAPKs (Appendix A) after pathogen treatment. In a different activation pathway from NF-κB, IRF3 is phosphorylated by internalizing TLR4 [39]. This was supported by an invasion assay, which showed that a bacterial MOI of 100 did not allow sufficient infiltration into host cells; after increasing the MOI to 200, bacteria were barely detected in PDLFs. Overall, our experimental condition using a bacterial MOI of 100 was insufficient to induce TLR4 internalization, but it was sufficient to stimulate TLR4-mediated NF-κB activation.

It is difficult to find natural products that are effective against periodontal inflammation. In our previous report on *Litsea japonica* leaf extract (*LJLE*), periodontal cells stimulated with *F. nucleatum* for 12 h expressed IL-6 and IL-8 mRNA. The mRNA levels were decreased upon treatment with 100 µg/mL *LJLE*. Although the *LJLE* reduced the IL-6 mRNA expression by 75% and IL-8 by 67%, the *OJE* used in this study caused a decrease of over 90% in IL-6 mRNA expression and 85% in IL-8 mRNA expression under the same experimental conditions. In addition, the crude *OJE* was fractionated and evaluated to determine the fractions that contained effective substances. The EtOAc and CH_2_Cl_2_ fractions effectively reduced IL-6 and IL-8 expression in response to LPS or oral bacteria. There are several reports that the EtOAc fraction of *O. japonica* rhizome contains acetone-type phenolics that inhibit the production of NO, prostaglandin E2, and immune mediators, including TNF-α, IL-1β, and IL-6 [20,21]. Although we did not purify specific compounds from the extract, *O. japonica* extract was shown to have a meaningful anti-inflammatory effect on periodontitis. 

## 4. Materials and Methods

### 4.1. Ethics Statement

Chonnam National University Dental Hospital Institutional Review Board (Approval No., CNUDH-2016-014; 21 November 2016) approved the isolation of human PDLFs. Written informed consent was obtained from all subjects after the nature and possible outcomes of this study were described. 

### 4.2. Preparation of 70% Extract and Solvent Fractions from O. japonica

*Osmunda japonica* was collected at Sillye, Namwon, Jeju, Republic of Korea, in May 2015. A voucher specimen (No. JBRI-381) was deposited in Biodiversity Research Institute, Jeju Technopark. Air-dried *O. japonica* (100 g) was extracted with 1 L 70% EtOH for 24 h. The 70% EtOH extracts were combined and evaporated under reduced pressure to concentrate the extract; they were then sequentially partitioned with n-hexane, CH_2_Cl_2_, EtOAc, BuOH, and H_2_O (Scheme 1).

### 4.3. Bacterial Strains and Culture Condition

The *F. nucleatum* subspecies polymorphum American Type Culture Collection (ATCC) 10953T was purchased from ATCC (Manassas, VA, USA), and *P. gingivalis* KCOM 2804 was provided by the Korean Collection for Oral Microbiology (Gwangju, Korea). For the infection, the bacteria strains were prepared by culturing *F. nucleatum* and *P. gingivalis* in tryptic soy broth supplemented with 0.5% yeast extract, 0.05% cysteine-HCl-H_2_O, 10 µg/mL hemin, and 2 µg/mL vitamin K1. These strains were cultured under anaerobic conditions (10% H_2_, 5% CO_2_, and 85% N_2_) in a 37 °C anaerobic chamber (Bactron I; Sheldon Manufacturing Inc., Cornelius, OR, USA). The number of bacteria was counted by measuring the absorbance of cultured bacteria at 600 nm, which corresponds to 1 × 10^8^ bacteria/mL. The bacteria were washed with D-PBS before infection, suspended in Dulbecco’s modified Eagle’s medium (DMEM), and treated to PDLFs at a multiplicity of infection (MOI) of 100.

### 4.4. Primary PDLFs

Primary human PDLFs were prepared as described by Somerman et al. [40]. All participants were healthy adults without periodontal disease. The PDLFs were grown in DMEM (Gibco BRL, Waltham, MA, USA) supplemented with 10% heat-inactivated fetal bovine serum (GenDepot, Katy, TX, USA), 100 U/mL penicillin, and 100 µg/mL streptomycin (GenDepot, USA) at 37 °C in a humidified atmosphere containing 5% CO_2_. Cells were detached using 0.25% trypsin/0.02% ethylenediaminetetraacetic acid (Sigma-Aldrich, St. Louis, MO, USA), when they had reached confluence. The PDLFs were sub-cultured five to six times at a ratio of 1:2 to 1:3 in 100 mm culture dish and then used for the experiments.

### 4.5. Measurement of Cell Viability

We used the 3-(4,5-dimethylthiazol-2-yl)-2,5-diphenyltetrazolium bromide (MTT) colorimetric assay to measure cell viability after treatment with the *O. japonica* extract. PDLFs were seeded on 96-well plates at a density of 10^4^ cells/well in DMEM containing 10% fetal bovine serum. The cells were incubated with various concentrations of the extract for the indicated times; then, the medium was carefully replaced with fresh medium containing MTT (EZ-Cytox; DoGenBio, Gwangju, Korea) solution. Subsequently, the plates were incubated for 3 h at 37 °C in an atmosphere of 5% CO_2_. The absorbance at 450 nm was measured using a microplate reader (Bio-Rad Laboratories, Hercules, CA, USA). The reported values are the means of triplicate experiments.

### 4.6. Quantitative Real Time-PCR

Total RNA was isolated using RNeasy kits (Qiagen, Hilden, Germany). The isolated RNA was reverse transcribed using random hexamer oligonucleotides (Takara Biotechnology, Tokyo, Japan). Real-time quantitative PCR was performed using SYBR Green Master Mix (Takara Biotechnology, Tokyo, Japan). The following primers were used: human IL-6 F: 5′-agggctcttcggcaaatgta-3′ and R: 5′-tgcccagtggacaggtttc-3′; and human IL-8 F: 5′-tttctgttaaatctggcaaccctagt-3′ and R: 5′-ataaaggagaaaccaaggcacagt-3′. The expression of human glyceraldehyde 3-phosphate dehydrogenase (GAPDH) was used for data normalization. The primers for hGAPDH RT-PCR were: human GAPDH F: 5′-accccttcattgacctcaac-3′ and R: 5′- cttgacggtgccatctgaatt-3′.

### 4.7. ELISA

Concentrations of IL-6 and IL-8 in the secreted cell culture supernatants were quantified by using human-specific ELISA kits (Biolegend, San Diego, CA, USA). Briefly, 96-well plate was coated with anti-human IL-6 or IL-8 monoclonal antibodies. By incubating the coated plate with the assay solution for 2 h, non-specific binding was blocked. Then, 100 µL of the standard IL-6, IL-8, or culture supernatants were added for quantification. The cytokines were detected by using a horseradish peroxidase-labeled monoclonal antibody against each target protein. The plate was incubated for 2 h at room temperature and then washed to remove the unbound enzyme-labeled antibodies. The colorimetric reaction of horseradish peroxidase was determined by the addition of the substrate solution. The reaction was stopped by the addition of sulfuric acid. The absorbance at 450 nm was measured by using a SpectaMax i3X micro-titer plate reader (San Jose, CA, USA). The concentrations of each target were determined by interpolation from a standard curve and presented as pictograms per milliliter (pg/mL) ± one standard error of the mean.

### 4.8. Bacterial Invasion Assay

Invasion assays were conducted as previously described [33] with modifications for oral bacteria. PDLFs were plated at 2 × 10^5^ cells per well in a 24-well plate 1 day before infection. The cells were infected with *F. nucleatum* or *P. gingivalis* for 2 h at an MOI of 100. Non-adherent bacteria were removed by three washes with PBS; subsequently, gentamicin (300 µg/mL) and metronidazole (200 µg/mL; Sigma) incorporated medium was added, and the monolayers were incubated for 1 h before lysis using 1 mL double-distilled water. The lysates were serially diluted 10-fold with PBS; then, aliquots were plated on tryptic soy agar (as described above) and incubated under anaerobic conditions for 48–72 h, and the surviving bacteria were counted.

### 4.9. Statistical Analysis

We performed statistical analysis using one-way analysis of variance (ANOVA) with Tukey’s test, and the analysis was computed by using the Statistical Packages for Social Science (SPSS, 23.0, Chicago, IL, USA). P-values less than 0.05 were considered statistically significant and are indicated by asterisks in the figures. Values are presented as the mean ± SD (indicated by error bars) of three independent experiments performed in triplicate. 

### 4.10. Western Blotting Analysis

PDLFs in culture were detached by scraping and treated with RIPA buffer containing a protease inhibitor cocktail and protein phosphatase inhibitors. The cell lysate was incubated on ice for 30 min and centrifuged; then, 20 µg of the supernatant was separated by SDS-PAGE and transferred to a PVDF membrane. Protein expression was measured using NF-kB or phospho-NF-kB antibodies purchased from Cell Signaling Technology (Boston, MA, USA) and normalized to the expression of mouse monoclonal β-actin antibody (Sigma-Aldrich, St Louis, MO, USA).

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
