# Peer review of "Osmunda japonica* Extract Suppresses Pro-Inflammatory Cytokines by Downregulating NF-κB Activation in Periodontal Ligament Fibroblasts Infected with Oral Pathogenic Bacteria"

_ijms, 2020, doi:10.3390/ijms21072453_

Round 1
Reviewer 1 Report
Manuscript submitted for revision is written very well with no spelling mistakes. In vitro experiments are interesting and worth of transfer to in vivo tests. A few problems (citations from the text are itallic) should be explained:
Line 50
Periodontal ligaments are the connective tissues located between the cementum of the teeth and the
50 alveolar bone of the mandible.
Periodontal ligaments exist also between teeth and maxilla.
Line 89
In the Figure 1 description „NT” symbol should be defined.
Line 122
Periodontal inflammation is caused by mixed bacterial flora. Two choosen bacteria species are the most malicious. To investigate the real state samples of bacteria should be taken from patients’ periodontal tissue.
Line 165,166
The ethyl acetate fraction was cytotoxic at a
165 concentrations over 100 μg/mL (Figure 5A). Therefore, PDLFs were treated with 50 μg/mL of each
166 fraction and stimulated with ecLPS to evaluate the anti-inflammatory effect of each fraction.
Only ethyl acetate fraction was toxic in a higher concentration. Why the concentrations of other fractions was decreased?
Line 162
Butanol fraction should be mentioned.
Line 267
Air-dried
267 O. japonica (100 g) was extracted with 1 mL 70% EtOH for 24 h. The 70% EtOH extracts were combined
The volume of extractant is too smal for 100g of the herb. If the extrats were combined more detailed description of extract preparation should be given. Was it exhaustive with one sample of herb or combined from many samples?
Line 268, 269
268 and evaporated under reduced pressure to concentrate the extract, and then sequentially partitioned
269 with n-hexane, CH2Cl2, EtOAc, BuOH, and H2O (Table1).
Why the extract was partitioned after first extraction? Most unpolar compounds were not exreacted from the herb by 70% ethanol. To have unpolar compounds fraction for analyse n-hexane should be used to extract sample of herb directly. Similar situation is for dichloromethane. Why the herb was not extracted with use all solvents separately?
Line 406
There is a lack of space in the article title.
Line 425,426
In bibliographic description the year of publication should be precised.
Author Response
Dear Reviewer #1, We are grateful for your careful review of our manuscript and for providing us with comments and suggestions to improve the quality of the manuscript.
Q1. Line 50 Periodontal ligaments are the connective tissues located between the cementum of the teeth and the alveolar bone of the mandible.
Periodontal ligaments exist also between teeth and maxilla.
A1 . We will change the sentence to ‘Periodontal ligaments are the connective tissue generally located between the cementum of the teeth and the alveolar bone of the mandible or maxilla.’
Q2. Line 89 In the Figure 1 description „NT” symbol should be defined.
A2. We missed this annotation. We will include ‘NT: no treatment’
Q3. Line 122 Periodontal inflammation is caused by mixed bacterial flora. Two choosen bacteria species are the most malicious. To investigate the real state, samples of bacteria should be taken from patients’ periodontal tissue.
A3. Although we did not use bacteria from patients, we attempted to mimic the conditions of severe periodontitis. To simplify the experiment, we selected two malicious bacteria; they are type strains and have been used in our previous reports [1-3]. One is F. nucleatum, which produces anaerobic conditions. The other is P. gingivalis, which is found in deep periodontal pockets. The anaerobic conditions produced by F. nucleatum provide a suitable environment for P. gingivalis to grow rapidly and exhibit virulence. For these reasons, we chose these bacteria to perform the experiment.
Q4. Line 165,166 The ethyl acetate fraction was cytotoxic at a 165 concentrations over 100 μg/mL (Figure 5A). Therefore, PDLFs were treated with 50 μg/mL of each fraction and stimulated with ecLPS to evaluate the anti-inflammatory effect of each fraction.
Only ethyl acetate fraction was toxic in a higher concentration. Why the concentrations of other fractions was decreased?
A4. As we do not know the specific compounds in each partition and we wish to synchronize the quantity of input materials for the comparison of effective partition, we chose the highest concentration that did not show toxicity in all partitions.
Q5. Line 162 Butanol fraction should be mentioned.
A5. We have included the word ‘butanol.’
Q6. Line 267 Air-dried O. japonica (100 g) was extracted with 1 mL 70% EtOH for 24 h. The 70% EtOH extracts were combined
The volume of extractant is too small for 100g of the herb. If the extracts were combined more detailed description of extract preparation should be given. Was it exhaustive with one sample of herb or combined from many samples?
A6. Usually the extraction is performed using 10 volumes of solvent. Hence, we revised the phrase ‘1 mL’ to ‘1 L’.
Q7. Line 268, 269 and evaporated under reduced pressure to concentrate the extract, and then sequentially partitioned with n-hexane, CH2Cl2, EtOAc, BuOH, and H2O (Table1).
Why the extract was partitioned after first extraction? Most unpolar compounds were not extracted from the herb by 70% ethanol. To have unpolar compounds fraction for analyze n-hexane should be used to extract sample of herb directly. Similar situation is for dichloromethane. Why the herb was not extracted with use all solvents separately?
A7. The fractionation method is used to identify effective or indicator components. In the study of natural products, fractionation is a method of separating components depending on the polarity, although each solvent may extract only a specific substance. After the extract is made turbid in H2O, if hexane → CH2Cl2 → EtOAc → BuOH → residues are used to obtain fractions using a polar solvent, polar substances can be extracted. In addition, we changed ‘table 1’ to ‘scheme 1’ to show the schematic process of fractionation and extraction.
Q8. Line 406 There is a lack of space in the article title.
A8. We have added a space.
Q9. Line 425,426 In bibliographic description the year of publication should be precised.
A9. We have confirmed the journal information. The precise name of the journal is ‘Periodontology 2000’, and the paper was published in 1994 (Online ISSN 1600-0757). We have re-written the reference information correctly.
[A1]As you have included a general statement to reviewer 1 to thank them for their comments, I have removed expressions of gratitude from each individual answer.
[A2]Would “dissolved in” or “suspended in” be more appropriate here?
Reviewer 2 Report
Comments and Suggestions for Authors:
In this paper, the authors investigated the anti-inflammatory effects on primary PDLFs stimulated by oral pathogens of Osmunda japonica extract. However, this paper have some problems that should be explained.
- Section 2.3, the bacterial invasion assay (4C) was performed with nucleatum single or mixed strains. But, the standard deviation is too large, and the data needs to be tested again.
- Section 2.4, why the cytotoxicity of the BuOH fraction is not shown in fig 5A? In Figure 5B, the relative expression of IL-6 mRNA was also significantly reduced after treatment with the BuOH fraction. Based on the yield of each fractions, the amount of BuOH fraction (16.59%) was much larger than CH2Cl2 fraction 6.72% and EA fraction 4.87%. BuOH fraction also seems to be one of the important fraction for reducing the expression of proinflammatory cytokines. It is best to add the activity of BuOH fraction in IL-6 and IL-8 secretion experiments (Fig 5D and 5E).
- Section 2.2, line 116-117, “the pre-treatment of OJE significantly reduced IL-6 (Figure 3A–3C) and IL-8 (Figure 3D–3F) secretion…” and line 118-119 “The gingivalis infection induced relatively lower cytokine expression than F. nucleatum…” The description in this paragraph does not clearly express how much the activity changes. Authors can use percentages (%) to represent changes and make them easier to understand.
- Line 267, “ japonica (100 g) was extracted with 1 mL 70% EtOH” I don’t think 100g of plant material can be extracted by 1 mL solvent, please checked the preparation process again.
Author Response
Dear Reviewer, We thank you for your careful review of our manuscript and for providing us with your comments and suggestions to improve the quality of the manuscript.
Q1. Section 2.3, the bacterial invasion assay (4C) was performed with nucleatum single or mixed strains. But, the standard deviation is too large, and the data needs to be tested again.
A1. We have added the P-values that show that there was no significant difference between OJE treated and no treatment. Thus, Fig 4 indicates that the OJE treatment did not affect bacterial internalization into host cells.
Q2. Section 2.4, why the cytotoxicity of the BuOH fraction is not shown in fig 5A? In Figure 5B, the relative expression of IL-6 mRNA was also significantly reduced after treatment with the BuOH fraction. Based on the yield of each fractions, the amount of BuOH fraction (16.59%) was much larger than CH2Cl2 fraction 6.72% and EA fraction 4.87%. BuOH fraction also seems to be one of the important fraction for reducing the expression of proinflammatory cytokines. It is best to add the activity of BuOH fraction in IL-6 and IL-8 secretion experiments (Fig 5D and 5E).
A2. We have added the revised Figure 5A, including the BuOH and hexane results. The BuOH fraction was not cytotoxic, but we excluded it from the original data. As the fraction has lots of pigments, they showed higher optical density after the concentration was increased. We also added the following sentence to provide a better explanation: ‘At the mRNA level, the ethyl acetate and CH2Cl2 fractions showed the best effect on the reduction of IL-6 and IL-8 mRNA expression (Fig 5B and 5C). Therefore, we chose to examine the protein expression of these two fractions (Fig 5D and 5E).’
Q3. Section 2.2, line 116-117, “the pre-treatment of OJE significantly reduced IL-6 (Figure 3A–3C) and IL-8 (Figure 3D–3F) secretion…” and line 118-119 “The gingivalis infection induced relatively lower cytokine expression than F. nucleatum…” The description in this paragraph does not clearly express how much the activity changes. Authors can use percentages (%) to represent changes and make them easier to understand.
A3. We added the sentence ‘Compared with F. nucleatum infection (at 12 h, without OJE), the expression of IL-6 and IL-8 in P. gingivalis-infected cells was 90% and 82% lower, respectively.’
Q4. Line 267, “ japonica (100 g) was extracted with 1 mL 70% EtOH” I don’t think 100g of plant material can be extracted by 1 mL solvent, please checked the preparation process again.
A4. Usually the extraction is performed using 10-fold volume of solvent. Therefore, we revised the phrase ‘1 mL’ to ‘1 L’.
Round 2
Reviewer 2 Report
Comments and Suggestions for Authors:
Most of the comments have been answered, the manuscript should be accepted for publication in this form.